# LEARN2AGREE: FITTING WITH MULTIPLE ANNOTATORS WITHOUT OBJECTIVE GROUND TRUTH

**Chongyang Wang**[1], **Yuan Gao**[2], **Chenyou Fan**[3], **Junjie Hu**[2], **Tin Lum Lam**[2]
**Nicholas D. Lane**[4], **Nadia Bianchi-Berthouze**[5]
Tsinghua University[1]
Shenzhen Institute of Artificial Intelligence and Robotics for Society[2]
South China Normal University[3], University of Cambridge[4], University College London[5]
`wangchongyang@tsinghua.edu.cn`, {`gaoyuankid, fanchenyou`}`@gmail.com`
{`hujunjie, tllam`}`@cuhk.edu.cn`, `ndl32@cam.ac.uk`
`nadia.berthouze@ucl.ac.uk`

## ABSTRACT

The annotation of domain experts is important for some medical applications where the objective ground truth is ambiguous to define, e.g., the rehabilitation for some chronic diseases, and the prescreening of some musculoskeletal abnormalities without further medical examinations. However, improper uses of the annotations may hinder developing reliable models. On one hand, forcing the use of a single ground truth generated from multiple annotations is less informative for the modeling. On the other hand, feeding the model with all the annotations without proper regularization is noisy given existing disagreements. For such issues, we propose a novel Learning to Agreement (Learn2Agree) framework to tackle the challenge of learning from multiple annotators without objective ground truth. The framework has two streams, with one stream fitting with the multiple annotators and the other stream learning agreement information between annotators. In particular, the agreement learning stream produces regularization information to the classifier stream, tuning its decision to be better in line with the agreement between annotators. The proposed method can be easily added to existing backbones, with experiments on two medical datasets showed better agreement levels with annotators.

## 1 INTRODUCTION

There exist difficulties for model development in applications where the objective ground truth is difficult to establish or ambiguous merely given the input data itself. That is, the decision-making, *i.e.* the detection, classification, and segmentation process, is based on not only the presented data but also the expertise or experiences of the annotator. However, the disagreements existed in the annotations hinder the definition of a good single ground truth. Therefore, an important part of supervise learning for such applications is to achieve better fitting with annotators. In this learning scenario, the input normally comprises pairs of $(\boldsymbol{X}_i, r_i^j)$, where $\boldsymbol{X}_i$ and $r_i^j$ are respectively the data of $i$-th sample and the label provided by $j$-th annotator. Given such input, naïve methods aim to provide a single set of ground truth label for model development. Therein, a common practice is to aggregate these multiple annotations with majority voting (Surowiecki, 2005). However, majority-voting could misrepresent the data instances where the disagreement between different annotators is high. This is particularly harmful for applications where differences in expertise or experiences exist in annotators.

Except for majority-voting, some have tried to estimate the ground truth label using STAPLE (Warfield et al., 2004) based on Expectation-Maximization (EM) algorithms. Nevertheless, such methods are sensitive to the variance in annotations and the data size (Lampert et al., 2016; Karimi et al., 2020). When the number of annotations per $\boldsymbol{X}_i$ is modest, efforts are put into creating models that utilize all the annotations with multi-score learning (Meng et al., 2011) or soft labels (Hu et al., 2016). Recent approaches have instead focused on leveraging or learning the expertise of annotators

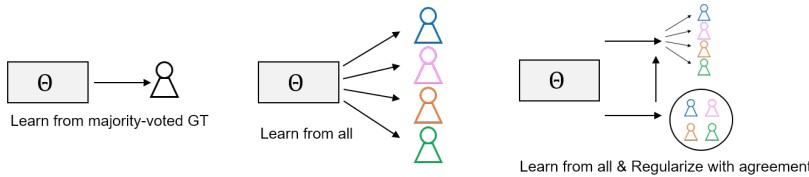

Figure 1: The proposed Learn2Agree framework regularizes the classifier that fits with all annotators with the estimated agreement information between annotators.

while training the model (Long et al., 2013; Long & Hua, 2015; Healey, 2011; Guan et al., 2018; Ji et al., 2021; Yan et al., 2014; 2010; Tanno et al., 2019; Zhang et al., 2020). A basic idea is to refine the classification or segmentation toward the underlying ground truth by modeling annotators.

In this paper, we focus on a hard situation when the ground truth is ambiguous to define. On one hand, this could be due to the missing of objective ground truth in a specific scenario. For instance, in the analysis of bodily movement behavior for chronic-pain (CP) rehabilitation, the self-awareness of people with CP about their exhibited pain or fear-related behaviors is low, thus physiotherapists play a key role in judging it (Felipe et al., 2015; Singh et al., 2016). However, since the physiotherapists are assessing the behavior on the basis of visual observations, they may disagree on the judgment or ground truth. Additionally, the ground truth could be temporarily missing, at a special stage of the task. For example, in abnormality prescreening for bone X-rays, except for abnormalities like fractures and hardware implantation that are obvious and deterministic, other types like degenerative diseases and miscellaneous abnormalities are mainly diagnosed with further medical examinations (Rajpurkar et al., 2017). That is, at prescreening stage, the opinion of the doctor makes the decision, which could disagree with other doctors or the final medical examination though.

Thereon, unlike the traditional modeling goal that usually requires the existence of a set of ground truth labels to evaluate the performance, the objective of modeling in this paper is to improve the overall agreement between the model and annotators. Our contributions are four-fold: (i) We propose a novel Learn2Agree framework to directly leverage the agreement information stored in annotations from multiple annotators to regularize the behavior of the classifier that learns from them; (ii) To improve the robustness, we propose a general agreement distribution and an agreement regression loss to model the uncertainty in annotations; (iii) To regularize the classifier, we propose a regularization function to tune the classifier to better agree with all annotators; (iv) Our method noticeably improves existing backbones for better agreement levels with all annotators on classification tasks in two medical datasets, involving data of body movement sequences and bone X-rays.

## 2 RELATED WORK

### 2.1 MODELING ANNOTATORS

The leveraging or learning of annotators' expertise for better modeling is usually implemented in a two-step or multiphase manner, or integrated to run simultaneously. For the first category, one way to acquire the expertise is by referring to the prior knowledge about the annotation, e.g. the year of experience of each annotator, and the discussion held on the disagreed annotations. With such prior knowledge, studies in Long et al. (2013); Long & Hua (2015); Healey (2011) propose to distill the annotations, deciding which annotator to trust for disagreed samples. Without the access to such prior knowledge, the expertise, or behavior of an annotator can also be modeled given the annotation and the data, which could be used as a way to weight each annotator in the training of a classification model Guan et al. (2018), or adopted to refine the segmentation learned from multiple annotators Ji et al. (2021). More close to ours are the ones that simultaneously model the expertise of annotators while training the classifier. Previous efforts are seen on using probabilistic models Yan et al. (2014; 2010) driven by EM algorithms, and multi-head models that directly model annotators as confusion matrices estimated in comparison with the underlying ground truth Tanno et al. (2019); Zhang et al. (2020). While the idea behind these works may indeed work for applications where the distance between each annotator and the underlying ground truth exists and can be estimated in some ways

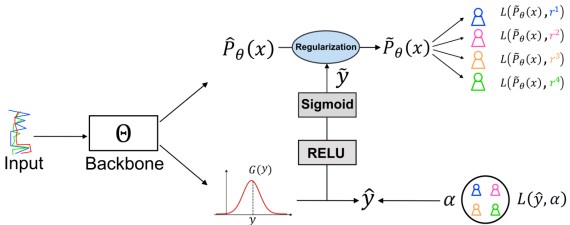

Figure 2: An overview of our Learn2Agree framework, comprising i) (above) the classifier stream with original prediction $\hat{p}_\theta(x_i)$ that fits with available annotations $\{r_i^j\}^{j=1,\ldots,J}$; and ii) (below) the agreement learning stream that learns to estimate $\hat{y}_i$ of the agreement level $\alpha_i$ between annotators, and leverage such information to compute the regularized prediction $\tilde{p}_\theta(x_i)$.

to refine the decision-making of a model, we argue that in some cases it is (at least temporarily) difficult to assume the existence of the underlying ground truth.

## 2.2 Modeling Uncertainty

Modeling uncertainty is a popular topic in the computer vision domain, especially for tasks of semantic segmentation and object detection. Therein, methods proposed can be categorized into two groups: i) the Bayesian methods, where parameters of the posterior distribution (e.g. mean and variance) of the uncertainty are estimated with Monte Carlo dropout Leibig et al. (2017); Kendall et al. (2017); Ma et al. (2017) and parametric learning Hu et al. (2020); Charpentier et al. (2020) etc.; and ii) 'non-Bayesian' alternatives, where the distribution of uncertainty is learned with ensemble methods Lakshminarayanan et al. (2016), variance propagation Postels et al. (2019), and knowledge distillation Shen et al. (2021) etc. Except for their complex and time-consuming training or inference strategies, another characteristic of these methods is the dependence on Gaussian or Dirac delta distributions as the prior assumption.

## 2.3 Evaluation without Ground Truth

In the context of modeling multiple annotations without ground truth, typical evaluation measures rely on metrics of agreements. For example, Kleinsmith et al. (2011) uses metrics of agreement, e.g. Cohen's Kappa Cohen (1960) and Fleiss' Kappa Fleiss (1971), as the way to compare the agreement level between a system and an annotator and the agreement level between other unseen annotators, in a cross-validation manner. However, this method does not consider how to directly learn from all the annotators, and how to evaluate the performance of the model in this case. For this end, Lovchinsky et al. (2019) proposes a metric named discrepancy ratio. In short, the metric compares performances of the model-annotator vs. the annotator-annotator, where the performance can be computed as discrepancy e.g. with absolute error, or as agreement e.g. with Cohen's kappa. In this paper, we use the Cohen's kappa as the agreement calculator together with such a metric to evaluate the performance of our method. We refer to this metric as agreement ratio.

## 3 Method

An overview of our proposed Learn2Agree framework is shown in Fig.2. The core of our proposed method is to learn to estimate the agreement between different annotators based on their raw annotations, and simultaneously utilize the agreement-level estimation to regularize the training of the classification task. Therein, different components of the proposed method concern: the learning of agreement levels between annotators, and regularizing the classifier with such information. In testing or inference, the model estimates annotators' agreement level based on the current data input, which is then used to aid the classification.

In this paper, we consider a dataset comprising $N$ samples $\mathbf{X} = \{x_i\}_{i=1,\ldots,N}$, with each sample $x_i$ being an image or a timestep in a body movement data sequence. For each sample $x_i$, $r_i^j$ denotes the annotation provided by $j$-th annotator, with $\alpha_i \in [0,1]$ being the agreement computed between

annotators (see Appendix for details). For a binary task, $r_i^j \in \{0, 1\}$. With such dataset $\mathcal{D} = \{x_i, r_i^j\}_{i=1,...,N}^{j=1,...,J}$, the proposed method aims to improve the agreement level with all annotators. It should be noted that, for each sample $x_i$, the method does not expect the annotations to be available from all the $J$ annotators.

## 3.1 MODELING UNCERTAINTY IN AGREEMENT LEARNING

To enable a robust learning of the agreement between annotators, we consider modeling the uncertainty that could exist in the annotations. In our scenarios, the uncertainty comes from annotators' varying expertise exhibited in their annotations across different data samples, which may not follow specific prior distributions. We propose a general agreement distribution $G(y_i)$ for agreement learning (see the upper part of Fig.3). Therein, the distribution values are the possible agreement levels $y_i$ between annotators with a range of $[0, 1]$, which is further discretized into $\{y_i^0, y_i^1, ... y_i^{n-1}, y_i^n\}$ with a uniform interval of $1/n$, with $n$ being a tunable hyperparameter deciding how precise the learning is. The general agreement distribution has a property $\sum_{k=0}^{n} G(y_i^k) = 1$, which can be implemented with a softmax layer with $n + 1$ nodes. The predicted agreement $\hat{y}_i$ for regression can be computed as the weighted sum of all the distribution values

$$\hat{y}_i = \sum_{k=0}^{n} G(y_i^k) y_i^k. \tag{1}$$

For training the predicted agreement value $\hat{y}_i$ toward the target agreement $\alpha_i$, inspired by the effectiveness of quantile regression in understanding the property of conditional distribution Koenker & Hallock (2001); Hao et al. (2007); Fan et al. (2019), we propose a novel Agreement Regression (AR) loss defined by

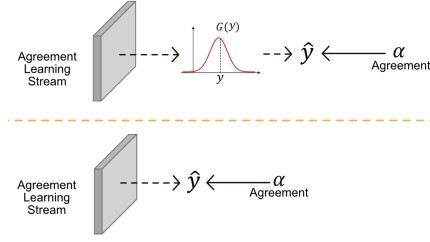

Figure 3: The learning of the agreement $\alpha_i$ between annotators is modeled with a general agreement distribution $G(y_i)$ using agreement regression loss $\mathcal{L}_{AR}$ (above), with the X axis of the distribution being the possible agreement levels $y_i$ and the Y axis being the respective probabilities. This learning can also be implemented as a linear regression task that learns to approach the exact agreement level $\alpha_i$ using RMSE loss (below).

$$\mathcal{L}_{AR}(\hat{y}_i, \alpha_i) = \max[\alpha_i(\hat{y}_i - \alpha_i), (\alpha_i - 1)(\hat{y}_i - \alpha_i)]. \tag{2}$$

Comparing with the original quantile regression loss, the quantile $q$ is replaced with the agreement $\alpha_i$ computed at current input sample $x_i$. The quantile $q$ is usually fixed for a dataset, as to understand the underlying distribution of the model's output at a given quantile. By replacing $q$ with $\alpha_i$, we optimize the general agreement distribution to focus on the given agreement level dynamically across samples.

In Li et al. (2021), the authors proposed to use the top $k$ values of the distribution and their mean to indicate the shape (flatness) of the distribution, which provides the level of uncertainty in object classification. In our case, all probabilities of the distribution are used to regularize the classifier. While this also informs the shape of the distribution for the perspective of uncertainty modeling, the skewness reflecting the high or low agreement level learned at the current data sample is also revealed. Thereon, two fully-connected layers with RELU and Sigmoid activations respectively are used to process such information and produce the agreement indicator $\tilde{y}_i$ for regularization.

### 3.1.1 LEARNING AGREEMENT WITH LINEAR REGRESSION.

Straightforwardly, we can also formulate the agreement learning as a plain linear regression task, modelled by a fully-connected layer with a Sigmoid activation function (see the lower part of Fig.3). Then, the predicted agreement $\hat{y}_i$ is directly taken as the agreement indicator $\tilde{y}_i$ for regularization. Given the predicted agreement $\hat{y}_i$ and target agreement $\alpha_i$ at each input sample $x_i$, by using Root Mean Squared Error (RMSE), the linear regression loss is computed as

$$\mathcal{L}_{RMSE}(\hat{y}, \alpha) = [\frac{1}{N} \sum_{i}^{N} (\hat{y}_i - \alpha_i)^2]^{\frac{1}{2}}. \tag{3}$$

It should be noted that, the proposed AR loss can also be used for this linear regression variant, which may help optimize the underlying distribution toward the given agreement level. In the experiments, an empirical comparison between different variants for agreement learning is conducted.

## 3.2 REGULARIZING THE CLASSIFIER WITH AGREEMENT

Since the high-level information implied by the agreement between annotators could provide extra hints in classification tasks, we utilize the agreement indicator $\tilde{y}_i$ to regularize the classifier training toward providing outcomes that are more in agreement with annotators. Given a binary classification task (a multi-class task can be decomposed into several binary ones), at input sample $x_i$, we denote the original predicted probability toward the positive class of the classifier to be $\hat{p}_\theta(x_i)$. The general idea is that, when the learned agreement indicator is i) at chance level i.e. $\tilde{y}_i = 0.5$, $\hat{p}_\theta(x_i)$ shall stay unchanged; ii) biased toward the positive/negative class, the value of $\hat{p}_\theta(x_i)$ shall be regularized toward the respective class. For these, we propose a novel regularization function written as

$$\tilde{p}_\theta(x_i) = \frac{\hat{p}_\theta(x_i)e^{\lambda(\tilde{y}_i-0.5)}}{\hat{p}_\theta(x_i)e^{\lambda(\tilde{y}_i-0.5)} + (1-\hat{p}_\theta(x_i))e^{\lambda(0.5-\tilde{y}_i)}}, \quad (4)$$

where $\tilde{p}_\theta(x_i)$ is the regularized probability toward the positive class of the current binary task, $\lambda$ is a hyperparameter controlling the scale at which the original predicted probability $\hat{p}_\theta(x_i)$ changes toward $\tilde{p}_\theta(x_i)$ when the agreement indicator increases/decreases. Fig.4 shows the

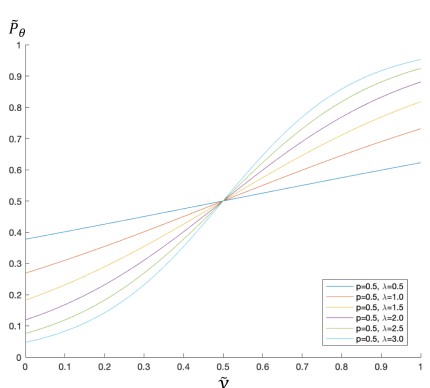

Figure 4: The property of the regularization function. X and Y axes are the agreement indicator $\tilde{y}_i$ and regularized probability $\tilde{p}_\theta(x_i)$, respectively. $\tilde{p}_\theta(x_i)$ is regularized to the class, for which the $\tilde{y}_i$ is high, with $\lambda$ controlling scale.

property of the function: for the original predicted probability $\hat{p}_\theta(x_i) = 0.5$, the function with larger $\lambda$ augments the effect of the learned agreement indicator $\tilde{y}_i$ so that the output $\tilde{p}_\theta(x_i)$ is regularized toward the more (dis)agreed; when $\tilde{y}_i$ is at 0.5, where annotators are unable to reach an above-chance opinion about the task, the regularized probability stays unchanged with $\tilde{p}_\theta(x_i) = \hat{p}_\theta(x_i)$.

## 3.3 COMBATING IMBALANCES IN LOGARITHMIC LOSS

In this subsection, we first alleviate the influence of class imbalances present in the annotation of each annotator, by refining the vanilla cross-entropy loss. We further explore the use of an agreement-oriented loss that may naturally avoid such imbalances during training.

### 3.3.1 ANNOTATION BALANCING FOR EACH ANNOTATOR.

For the classifier stream, given the regularized probability $\tilde{p}_\theta(x_i)$ at the current input sample $x_i$, the classifier is updated using the sum of the loss computed according to the available annotation $r_i^j$ from each annotator. Due to the various the nature of the task (*i.e.*, positive samples are sparse), the annotation from each annotator could be noticeably imbalanced. Toward this problem, we use the Focal Loss (FL) Lin et al. (2017), written as follows.

$$\mathcal{L}_{\mathrm{FL}}(p,g) = -|g-p|^\gamma (g\log(p) + (1-g)\log(1-p)), \quad (5)$$

where $p$ is the predicted probability of the model toward the positive class at the current data sample, $g \in \{0,1\}$ is the binary ground truth, and $\gamma \geq 0$ is the focusing parameter used to control the threshold for judging the well-classified. A larger $\gamma$ leads to a lower threshold so that more samples would be treated as the well-classified and down weighted. In our scenario, the FL function is integrated into the following loss function to compute the average loss from all annotators.

$$\mathcal{L}_\theta(\tilde{\mathbf{P}}_\theta, \mathbf{R}) = \frac{1}{J}\sum_{j=1}^{J}\frac{1}{\dot{N}^j}\sum_{i=1}^{\dot{N}^j}\mathcal{L}_{FL}(\tilde{p}_\theta(x_i), r_i^j), \quad (6)$$

where $\dot{N}^j \leq N$ is the number of samples that have been labelled by $j$-th annotator, $\tilde{\mathbf{P}}_\theta = \{\tilde{p}_\theta(x_i)\}_{i=1,\dots,N}$, $\mathbf{R} = \{r_i^j\}_{i=1,\dots,\dot{N}^j}^{j=1,\dots,J}$. $r_i^j = \text{null}$ if the $j$-th annotator did not annotate at $i-$th sample, and the loss is not computed here.

Additionally, searching for the $\gamma$ manually for each annotator could be cumbersome, especially for a dataset labeled by numerous annotators. In this paper, we compute $\gamma$ given the number of samples annotated by each annotator per class of each binary task. The hypothesis is that, for annotations biased more toward one class, $\gamma$ shall set to be bigger since larger number of samples tend to be well-classified. We leverage the effective number of samples Cui et al. (2019) to compute each $\gamma_j$ as follows.

$$\gamma_j = \frac{(1 - \beta^{n_k^j})}{(1 - \beta^{(\dot{N}^j - n_k^j)})}, \tag{7}$$

where $n_k^j$ is the number of samples for the majority class $k$ in the current binary task annotated by annotator $j$, $\beta = \frac{\dot{N}^j - 1}{\dot{N}^j}$.

### 3.3.2 AGREEMENT-ORIENTED LOSS.

In de La Torre et al. (2018), a Weighted Kappa Loss (WKL) was used to compute the agreement-oriented loss between the output of a model and the annotation of an annotator. As developed from the Cohen's Kappa, this loss may guide the model to pay attention to the overall agreement level instead of the local mistake. Thus, we may be able to avoid the cumbersome work of alleviating the class imbalances as above. This loss function can be written as follows.

$$\mathcal{L}_{\text{WKL}} = \log(1 - \kappa). \tag{8}$$

The linear weighted kappa $\kappa$ Cohen (1968) is used in this equation, where the penalization weight is proportional to the distance between the predicted and the class. We replace the FL loss written in Equation 5, to compute the weighted kappa loss across samples and annotators using Equation 6. The value range of this loss is $(-\infty, \log 2]$, thus a Sigmoid function is applied before we sum the loss from each annotator. We compare this WKL loss function to the logarithmic one in our experiment.

## 4 EXPERIMENTS

In this section, we evaluate our proposed method with data annotated by multiple human experts, where the objective ground truth is ambiguous to define. Please refer to the Appendix for dataset descriptions, implementation details, and the computation of agreement ground truth.

### 4.1 METRIC

Following Lovchinsky et al. (2019), we evaluate the performance of a model by using the agreement ratio as follows.

$$\Delta = \frac{\mathrm{C}_J^2}{J} \frac{\sum_{j=1}^J \mathrm{Sigmoid}(\kappa(\tilde{\mathbf{P}}_\theta, \mathbf{R}^j))}{\sum_{j,j'=1 \& j \neq j'}^J \mathrm{Sigmoid}(\kappa(\mathbf{R}^j, \mathbf{R}^{j'}))}, \tag{9}$$

where the numerator computes the average agreement for the pairs of predictions of the model and annotations of each annotator, and the denominator computes the average agreement between annotators with $\mathrm{C}_J^2$ denoting the number of different annotator pairs. $\kappa$ is the Cohen's Kappa. The agreement ratio $\Delta > 0$ is larger than 1 when the model performs better than the average annotator Lovchinsky et al. (2019).

### 4.2 RESULTS

#### 4.2.1 AGREEMENT-ORIENTED LOSS VS. LOGARITHMIC LOSS.

As shown in the first section of Table 1, models trained with majority-voted ground truth produce agreement ratios of 1.0417 and 0.7616 with logarithmic loss and annotation balancing (in this case is class balancing for the single majority-voted ground truth) on the EmoPain and MURA datasets,

Table 1: The ablation experiment on the EmoPain and MURA datasets. Majority-voting refers to the method using the majority-voted ground truth for training. CE and WKL refer to the logarithmic and weighted kappa loss functions used in the classifier stream, respectively. Linear and Distributional refer to the agreement learning stream with linear regression and general agreement distribution, respectively. The best performance in each section is marked in bold per dataset.

| Framework/Annotator | CE | WKL | Annotation Balance | Linear | Distributional | $\Delta\uparrow$ EmoPain | $\Delta\uparrow$ MURA |
|---|---|---|---|---|---|---|---|
| Majority Voting | √ | | √ | | | 1.0417 | 0.7616 |
|  | | √ | | | | **1.0452** | **0.7638** |
| Learn-from-all | √ | | | | | 0.9733 | 0.7564 |
|  | √ | | √ | | | 1.0189 | 0.7665 |
|  | | √ | | | | **1.0407** | **0.7751** |
| **Learn2Agree (Ours)** | √ | | √ | √ | | 1.0477 | 0.7727 |
|  | √ | | √ | | √ | 1.0508 | 0.7796 |
|  | | √ | | √ | | 1.0471 | 0.7768 |
|  | | √ | | | √ | **1.0547** | **0.7801** |
| Annotator 1 | | | | | | 0.9613 | **1.0679** |
| Annotator 2 | | | | | | 1.0231 | 0.9984 |
| Annotator 3 | | | | | | **1.0447** | 0.9743 |
| Annotator 4 | | | | | | 0.9732 | 0.9627 |

respectively. However, as shown in the second section of Table 1, directly exposing the model to all the annotations is harmful, with performances lower than the majority-voting ones of 0.9733 and 0.7564 achieved on the two datasets using logarithmic loss alone. By using the balancing method during training, the performance on the EmoPain dataset is improved to 1.0189 but is still lower than majority-voting one, while a better performance of 0.7665 than the majority-voting is achieved on the MURA dataset. These results show the importance of balancing for the modeling with logarithmic loss in a learn-from-all paradigm. With the WKL loss, performances of the model in majority-voting (1.0452/0.7638) and learn-from-all (1.0407/0.7751) paradigms are further improved. This shows the advantage of the WKL loss for improving the fitting with multiple annotators, which also alleviates the need to use class balancing strategies.

### 4.2.2 THE IMPACT OF OUR LEARN2AGREE METHOD.

For both datasets, as shown in the third section of Table 1, with our proposed Learn2Agree method using general agreement distribution, the best overall performances of 1.0547 and 0.7801 are achieved on the two datasets, respectively. For the agreement learning stream, the combination of general agreement distribution and AR loss shows better performance than its variant using linear regression and RMSE on both datasets (1.0477 with logarithmic loss and 0.7768 with WKL loss). Such results could be due to the fact that the agreement indicator $\tilde{y}_i$ produced from the linear regression is directly the estimated agreement value $\hat{y}_i$, which could be largely affected by the errors made during agreement learning. In contrast, with general agreement distribution, the information passed to the classifier is first the shape and skewness of the distribution $G(y_i)$. Thus, it is more tolerant to the errors (if) made by the weighted sum that used for regression with agreement learning.

### 4.2.3 COMPARING WITH ANNOTATORS.

In the last section of Table 1, the annotation of each annotator is used to compute the agreement ratio against the other annotators (Equation 9).

For the EmoPain dataset, the best method in majority-voting (1.0452) and learn-from-all (1.0407) paradigms show very competitive if not better performances than annotator 3 (1.0447) who has the best agreement level with all the other annotators. Thereon, the proposed Learn2Agree method improves the performance to an even higher agreement ratio of 1.0547 against all the annotators. This performance suggests that, when adopted in real-life, the model is able to analyze the protective behavior of people with CP at a performance that is highly in agreement with the human experts.

However, for the MURA dataset, the best performance so far achieved by the Learn2Agree method of 0.7801 is still lower than annotator 1. This suggests that, at the current task setting, the model may make around 22% errors more than the human experts. One reason could be largely due to the

Table 2: The experiment on analyzing the impact of Agreement Regression (AR) loss on agreement learning

| Dataset | Classifier Loss | Agreement Learning Type | Agreement Learning Loss | Δ ↑ |
|---|---|---|---|---|
| EmoPain | CE | Linear | RMSE | **1.0477** |
| | | | AR | 0.9976 |
| | | Distributional | RMSE | 1.0289 |
| | | | AR | **1.0508** |
| | WKL | Linear | RMSE | **1.0454** |
| | | | AR | 1.035 |
| | | Distributional | RMSE | 1.0454 |
| | | | AR | **1.0482** |
| MURA | CE | Linear | RMSE | **0.7727** |
| | | | AR | 0.7698 |
| | | Distributional | RMSE | 0.7729 |
| | | | AR | **0.7796** |
| | WKL | Linear | RMSE | **0.7707** |
| | | | AR | 0.7674 |
| | | Distributional | RMSE | 0.7724 |
| | | | AR | **0.7773** |

challenge of the task. As shown in Rajpurkar et al. (2017), where the same backbone only achieved a similar if not better performance than the other radiologists for only one (wrist) out of the seven upper extremity types. In this paper, the testing set comprises all the extremity types, which makes the experiment even more challenging. Future works may explore better backbones tackling this.

### 4.2.4 THE IMPACT OF AGREEMENT REGRESSION LOSS.

The proposed AR loss can be used for both the distributional and linear agreement learning stream. However, as seen in Table 2 and Table 2, the performance of linear agreement learning is better with RMSE loss rather than with the AR loss. The design of the AR loss assumes the loss computed for a given quantile is in accord with its counterpart of agreement level. Thus, such results may be due to the gap between the quantile of the underlying distribution of the linear regression and the targeted agreement level. Therefore, the resulting estimated agreement indicator using AR loss passed to the classifier may not reflect the actual agreement level. Instead, for linear regression, a vanilla loss like RMSE promises that the regression value is fitting toward the actual agreement level.

By contrast, the proposed general agreement distribution directly adopts the range of agreement levels to be the distribution values, which helps to narrow such a gap when AR loss is used. Therein, the agreement indicator is extracted from the shape and skewness of such distribution (probabilities of all distribution values), which could better reflect the agreement level when updated with AR loss. As shown, the combination of distributional agreement learning and AR loss achieves the best performance in each dataset.

## 5 CONCLUSION

In this paper, we targeted the scenario of learning with multiple annotators where the ground truth is ambiguous to define. Two medical datasets in this scenario were adopted for the evaluation. We showed that backbones developed with majority-voted *ground truth* or multiple annotations can be easily enhanced to achieve better agreement levels with annotators, by leveraging the underlying agreement information stored in the annotations. For agreement learning, our experiments demonstrate the advantage of learning with the proposed general agreement distribution and agreement regression loss, in comparison with other possible variants. Future works may extend this paper to prove its efficiency in datasets having multiple classes, as only binary tasks were considered in this paper. Additionally, the learning of annotator's expertise seen in Tanno et al. (2019); Zhang et al. (2020); Ji et al. (2021) could be leveraged to weight the agreement computation and learning proposed in our method for cases where annotators are treated differently.

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

ACKNOWLEDGMENTS

Chongyang Wang was supported by the Overseas Research Scholarship (ORS) and Graduate Research Scholarship (GRS) of University College London. This work was supported by the National Key R&D Program of China (2020YFB1313300).

## A  APPENDIX

### A.1  DATASETS

Two medical datasets are selected, involving data of body movement sequences and bone X-rays.

### A.1.1 EmoPain.

The EmoPain Aung et al. (2015) dataset contains skeleton-like movement data collected from 18 participants with CP and 12 healthy participants while they perform a variety of full-body physical rehabilitation activities (e.g. stretching forward and sitting down). In total, we have 46 activity sequences collected from these 30 participants, with each sequence lasting for about 10 minutes (or 36,000 samples). A binary task is included to detect the presence of protective behavior (e.g. hesitation, guarding) Keefe & Block (1982) exhibited by participants with CP during the performances. The detection of such behavior could be leveraged to generate automatic feedback and inform therapeutic personalized interventions Wang et al. (2021a). Four experts were recruited to provide the binary annotations of the presence or absence of protective behavior per timestep for each CP participant data sequence.

### A.1.2 MURA.

The MURA dataset Rajpurkar et al. (2017) comprises 40,561 radiographic images of 7 upper extremity types (i.e., shoulder, humerus, elbow, forearm, wrist, hand, and finger), and is used for the binary classification of abnormality. This dataset is officially split into training (36,808 images), validation (3197 images), and testing (556 images) sets, with no overlap in subjects. The training and validation sets are publicly available, with each image labelled by a radiologist. In the testing set, the authors of MURA recruited six additional radiologists for annotation, and defined the *ground truth* with majority-voting among three randomly-picked radiologists. The rest three radiologists achieved Cohen's kappa with such *ground truth* of 0.731, 0.763, and 0.778, respectively. To simulate the opinions of different experts for the data we have access to, three synthetic annotators are created to reach Cohen's kappa with the existing annotator at 0.80, 0.75, and 0.70, respectively.

### A.2 Implementation Details

For experiments on the EmoPain dataset, the state-of-the-art HAR-PBD network Wang et al. (2021a) is adopted as the backbone, and Leave-One-Subject-Out validation is conducted across the participants with CP. The average of the performances achieved on all the folds is reported. The training data is augmented by adding Gaussian noise and cropping, as seen in Wang et al. (2021b). The number of bins used in the general agreement distribution is set to 10, i.e., the respective softmax layer has 11 nodes. The $\lambda$ used in the regularization function is set to 3.0. For experiments on the MURA dataset, the Dense-169 network Huang et al. (2017) pretrained on the ImageNet dataset Deng et al. (2009) is used as the backbone. The original validation set is used as the testing set, where the first view (image) from each of the 7 upper extremity types of a subject is used. Images are all resized to be $224 \times 224$, while images in the training set are further augmented with random lateral inversions and rotations of up to 30 degrees. The number of bins is set to 5, and the $\lambda$ is set to 3.0. The setting of number of bins (namely, $n$ in the distribution) and $\lambda$ was found based on a grid search across their possible ranges, i.e., $n \in \{5, 10, 15, 20, 25, 30\}$ and $\lambda \in \{1.0, 1.5, 2.0, 2.5, 3.0, 3.5\}$.

For all the experiments, the classifier stream is implemented with a fully-connected layer using a Softmax activation with two output nodes for the binary classification task. Adam Kingma & Ba (2014) is used as the optimizer with a learning rate $lr =$ 1e-4, which is reduced by $1/10$ if the performance is not improved after 10 epochs. The number of epochs is set to 50. the logarithmic loss is adopted by default as written in Equation 5 and 6, while the WKL loss (8) is used for comparison when mentioned. For the agreement learning stream, the AR loss is used for its distributional variant, while the RMSE is used for its linear regression variant. We implement our method with TensorFlow deep learning library on a PC with a RTX 3080 GPU and 32 GB memory.

### A.3 Agreement Computation

For a binary task, the agreement level $\alpha_i$ between annotators is computed as follows.

$$\alpha_i = \frac{1}{\dot{J}} \sum_{j=1}^{\dot{J}} w_i^j r_i^j, \tag{10}$$

where $\dot{J}$ is the number of annotators that have labelled the sample $x_i$. In this way, $\alpha_i \in [0, 1]$ stands for the agreement of annotators toward the positive class of the current binary task. In this work,

we assume each sample was labelled by at least one annotator. $w_i^j$ is the weight for the annotation provided by $j$-th annotator that could be used to show the different levels of expertise of annotators. The weight can be set manually given prior knowledge about the annotator, or used as a learnable parameter for the model to estimate. In this work, we treat annotators equally by setting $w_i^j$ to 1. We leave the discussion on other situations to future works.

