# OpenReview forum: "Learn2Agree: Fitting with Multiple Annotators without Objective Ground Truth"
_ICLR.cc/2023/Workshop/TML4H — ICLR 2023 Workshop TML4H Poster_

### Official Review · Reviewer_EHpB · 2023-03-01

**Rating:** 7
**Confidence:** 4

**Review:**

#### Summary:

This paper proposes a novel Learning to Agreement (Learn2Argre) framework to tackle the challenge of learning from multiple annotators without object ground truth.

#### Strengths:
- This paper proposes a novel Learn2Agree framework, which learn to estimate the agreement between different annotators based on their raw annotations, and simultaneously utilize the agreement-level estimation to regularize the training of the classification task.
- The paper also discusses an agreement-oriented loss that may naturally avoid such imbalances during training.

#### Weakness:
- It is not clear that how to select the $n$ for $[y_i^0, y_i^n]$?
- The paper only discusses the application of the proposed framework in a binary classification problem. Could this framework be applied to the multi-class classification problem?
- It will be helpful to discuss how the $\lambda$ affects the performance of the model.

---

### Official Review · Reviewer_JzQx · 2023-03-01

**Rating:** 4
**Confidence:** 3

**Review:**

The paper studies the "annotation agreement learning" problem for utilizing ML models in medical applications.
The motivation is well established in the introduction, as annotation disagreements widely exist in medical ML tasks.
The authors propose a learning-based framework to regularize a classifier to fit with all annotations while estimating the agreement level between multiple annotators simultaneously.
However, the technical part needs major revision to make it accessible to the readers. I will put detailed comments below.

**Strengths**
- Clear motivation to demonstrate the importance of the studied problem
- Reasonable experiments section for validating the proposed framework

**Weaknesses/Recommended Changes**
- Some notations/terms need to be clarified:
  - Sec. 1, why does $r_i^j$ denote the i-th sample's label provided by r-th annotator? Should it be j-th annotator?
  - Sec. 3, how does $\alpha_i$ derive from the multiple annotations $r_i^j\ for\ j=0,\dots,J$?
  - Sec. 3, authors state "the agreement levels $y_i$ with a range of $[y_i^0, y_i^n]$". If $y_i$ is an prediction for the agreement $\alpha_i \in [0, 1]$, then how can $y_i$ be discretized into $\{y_i^0, \dots, y_i^n\}$ with interval of 1? Moreover, the Li et al. (2020) paper proposes the discretization technique to handle the bounding box representation, which makes sense to utilize such technology since the bounding box has a certain range. I would suggest instead of saying "inspired by Li et al. (2020)", it would be better to state the benefits of discretization.
- Given the proposed framework mainly follows the existing work, it shows limited novelty.

---

### Meta-Review · Area_Chair_H8Xv · 2023-03-05

**Recommendation:** Accept (Poster)
**Confidence:** 5

**Metareview:**

The paper proposed a learning to agree method to address the annotation disagreement in medical image, which is a common and crutial problem. Reviews agree that the paper studies an important problem, and experiments could strongly support the claims.
Considering the contribution, I would suggest accept this paper. However, upon acceptance, the authors are highly suggested to clarify the technichal details as pointed out by the reviews.